# Myocardial Infarction in Young Athletes

**DOI:** 10.3390/diagnostics13152473

**Published:** 2023-07-25

**Authors:** Mariusz Dotka, Łukasz A. Małek

**Affiliations:** 1Faculty of Medicine, Poznan University of Medical Sciences, 61-701 Poznań, Poland; mariusz.dotka7@gmail.com; 2Faculty of Rehabilitation, University of Physical Activity in Warsaw, 01-968 Warsaw, Poland

**Keywords:** myocardial infarction, young athletes, sudden cardiac death, mechanism, risk factor

## Abstract

Myocardial infarction (MI) in young athletes is very rare but can have serious consequences, including sudden cardiac death (SCD), an increased proarrhythmic burden in future life, and/or heart failure. We present two cases of young athletes with MI. They did not have previous symptoms, traditional risk factors, or a family history of MI. One case involves a 37-year-old male amateur athlete who experienced two MI following intense physical exertion, likely due to the erosion of an insignificant atherosclerotic plaque caused by a sudden increase in blood pressure during exercise. The second case describes a 36-year-old male semi-professional runner who collapsed at the finish line of a half-marathon and was diagnosed with hypertrophic cardiomyopathy. The heart’s oxygen demand–supply mismatch during intensive exercise led to MI. Following the case presentation, we discuss the most common causes of MI in young athletes and their mechanisms, including spontaneous coronary artery dissection, chest trauma, abnormalities of the coronary arteries, coronary artery spasm, plaque erosion, hypercoagulability, left ventricular hypertrophy, and anabolic steroids use.

## 1. Introduction

Myocardial infarction (MI) is characterized by the occurrence of acute myocardial damage caused by inadequate blood oxygen supply to the cardiac muscle, as indicated by the presence of abnormal cardiac biomarkers, in conjunction with evidence of acute myocardial ischemia [1]. It is assessed and diagnosed on the grounds of clinical evaluation, biochemical testing, electrocardiogram, as well as invasive and noninvasive imaging. Sometimes, MI can be confirmed only in the autopsy report [2]. The course and extent of myocardial infarction can vary. It can cause no symptoms and go undetected or can lead to serious complications and even sudden cardiac death (SCD) [3]. The majority of athletes who die from sudden cardiac death had no previous symptoms [4]. Another complication of MI in the longer term is a risk of severe ventricular arrhythmias and/or heart failure, which could also have a detrimental effect on an athlete’s career.

During competitive sports, the occurrence rate of sudden cardiac arrest (SCA) was recorded as 0.76 cases per 100,000 athlete-years. In contrast, the general population of the same age group has reported an incidence rate of SCA at 4.84 cases per 100,000 person-years [5]. Individuals who engage in any type of physical activity demonstrate a reverse dose-response association between the proportion of vigorous activity and mortality [6]. It has been also shown that the incidence of SCD is significantly higher in the group of individuals not associated with sports compared to those involved in recreational or competitive sports [7]. In groups unrelated to sport it was nearly four times higher compared to sports-related SCD (2.46 vs. 0.59). Therefore, regular physical activity has an overall beneficial effect on the risk of SCD and MI but the risk persists [7].

Out of all cases of sports-related SCD, 13.4% were attributed to acute MI, and these cases occurred rarely in individuals below 35–40 years of age; however, in the subgroup of young male competitive athletes, acute MI, despite being a rare event, is identified as one of the leading causes of SCD. Coronary artery disease, with or without acute MI, is identified as the most common underlying pathology in the group of individuals not associated with sports or involved in recreational or competitive sports [7]. Another study has shown that in athletes below the age of 35, the most common cause of SCA was idiopathic ventricular fibrillation, accounting for 62.5% (five individuals), while acute MI accounted for only 12.5% (one individual) [8].

In young athletes under 35–40 years old, the occurrence of MI type I, which is typically associated with atherosclerosis resulting from risk factors such as obesity, diabetes, smoking, or dyslipidemia, is only one of the mechanisms [9,10]. Many MIs are of the type II mechanism caused by oxygen supply–demand mismatch from abnormalities of the coronary arteries, spontaneous coronary artery dissection, coronary artery spasm, hypercoagulability, or consequences of anabolic steroids use.

The healing process of tissue after MI occurs in three stages: inflammatory, proliferative/reparative, and maturation. Both the excessive inflammatory response and excessive fibrosis can have proarrhythmic effects, and may especially contribute to malignant ventricular arrhythmias. The compact scar consists mainly of non-cellular tissue and lacks electrical excitability; however, it can act as an isolated region where reentrant arrhythmia can become anchored and develop into sustained ventricular tachycardia [11]. Ventricular fibrillation or tachycardia can occur primarily, within the first 4 h of ischemia, as well as secondarily, due to the remodeling or scar formation after 48 h of myocardial infarction [12].

This review aims to present and discuss potential causes of MI in young athletes to improve screening to identify athletes at risk and to facilitate management and treatment to reduce the risk of relevant complications. In this review, we will examine the available literature including clinical studies, case reports, and case series, to provide a comprehensive understanding of the unique aspects of MI in the population of young athletes (below 35–40 years of age).

## 2. Case Reports

### 2.1. Case 1

A 37-year-old male amateur athlete was admitted to the hospital due to severe chest and back pain. He had performed strenuous swimming training a day before the presentation. His ECG was unremarkable but biochemical examination revealed a markedly elevated concentration of troponin I, atypical for the exercise (>5 times the upper reference limit). For this reason, he was referred for a coronary angiogram but no changes were found. The pain subsided after a few hours. An MI with non-obstructed coronary arteries (MINOCA) was diagnosed and the patient was sent for cardiac magnetic resonance (CMR), which showed the presence of edema and small subendocardial late gadolinium enhancement (LGE) in the interventricular septum, typical for recent myocardial infarction in the area of the right coronary artery (RCA) supply. A thorough examination of the coronary angiogram demonstrated a lower opacification of one of the branches of RCA (Figure 1A). The patient was given a small dose of aspirin, high-dose of statin, and angiotensin convertase inhibitor (ACE-I). He had no hyperlipidemia, normal blood pressure, normal body weight, followed a healthy diet, and was not a smoker. He also did not have any family history of myocardial infarctions. However, he admitted to living under a great amount of psychological stress. After a few months, he resigned from medications and returned to physical activity. A year later he was admitted to the hospital again with similar symptoms, only this time after running. There were visible ischemic changes in the ECG in the form of a T-wave inversion in leads V4–V5 (Figure 1B) but again no changes in the coronary angiogram. The second CMR disclosed the normal size of the heart chambers, with preserved global left ventricular systolic function (LVEF = 57%) but with apical akinesia and hypokinesia of the apical segment of the septum and partly of the middle infero-septal segment of the left ventricle. Therefore apart from the prior MI, there were also features of the new, healing transmural left ventricular apical infarction (Figure 1C,D). This time, the patient was instructed to remain on aspirin, statin, and a small dose of ACE-I and to avoid strenuous exercises. His LDL cholesterol concentration remained below 55 mg%. It is hard to unanimously establish the cause of MI in the presented case; however, based on the available examinations, it was caused most likely by the erosion of a small, but unstable, atherosclerotic plaque. The incidents could have been provoked by the sudden rise in blood pressure with or without coronary artery spasm during a bout of intensive exercise. As intravascular imaging was not performed and would be probably impossible given the small diameter of the affected arteries, a final etiology could not be found. The patient refused to undergo computed tomography of the coronary arteries (CTCA) to analyze the presence of coronary calcifications and plaques. The patient has remained free of sequels for 2 years from the second event.

### 2.2. Case 2

A 36-year-old male semi-professional runner was referred for examination after collapsing at the finish of a half-marathon run. He had a 15-year-long history of long-distance running without symptoms. His ECG revealed signs of left ventricular hypertrophy (LVH) with left atrial enlargement and T-wave inversions in leads II, III, aVF, V3–V6 with an ST-segment depression of 1 mm in leads II, III, and aVF (Figure 2A). Transthoracic echocardiography revealed asymmetric hypertrophy of the interventricular septum of 17 mm suggestive of hypertrophic cardiomyopathy (HCM). The patient had no family history of HCM. CMR confirmed the diagnosis, showing also a mild left ventricular outflow tract obstruction (LVOTO) with systolic anterior movement (SAM) of the anterior mitral valve leaflet. There were no signs of non-ischemic LGE but there was a small subendocardial scar in the hypertrophied interventricular septum (Figure 2B,C), typical for prior small myocardial infarction. The patient had no signs of coronary artery disease on the CTCA. Other possible causes of secondary cardiac hypertrophy have been excluded, as per the guidelines on the diagnosis and treatment of hypertrophic cardiomyopathy [13]. Therefore, the MI was most probably attributable to a mismatch between oxygen demand and supply during intensive exercise. The patient was advised to refrain from competitive, strenuous exercises and was put on a small dose of beta-blocker.

## 3. Causes of MI in Young Athletes—The Literature Review

Below we present the literature review of the most frequent causes leading to MI in young athletes. Figure 3 presents the potential mechanisms connecting physical activity to the presented causes of MI in young athletes.

### 3.1. Spontaneous Coronary Artery Dissection (SCAD)

Several cases of MI associated with coronary artery dissection in young athletes have been described. It is a non-traumatic and non-iatrogenic spontaneous separation of the coronary artery wall [10,14,15]. Coronary artery dissection is more commonly observed in women [16]. A recent study has found that nearly 32% of participants reported engaging in very intense or unusual physical exertion in the two weeks before the occurrence of coronary artery dissection [16]. Coronary artery dissection may not be visible on coronary angiography [14]. The findings on an angiogram can resemble those seen in acute atherosclerotic plaque disease, coronary artery spasms, coronary thromboembolism, or even normal coronary arteries; however, advanced imaging techniques such as intravascular ultrasound and optical coherence tomography provide higher resolution, enabling more precise identification of the dissection flap and the accompanying intramural hematoma [10]. These techniques are recommended in the diagnosis of spontaneous coronary artery dissection [10]. Thanks to advances in technology, coronary CTCA plays an increasingly important role in the diagnosis of SCAD; however, the precise criteria for the diagnosis of SCAD using CTCA, as well as the sensitivity and specificity of such a test, have not yet been established [17,18]. The presence of a thrombus attached to the vessel wall, lack of atherosclerosis, and the history of extreme physical exertion leading to elevated forces exerted on the vessel wall may indicate the possibility of spontaneous coronary artery dissection, even despite the absence of typical criteria on optical coherence tomography imaging [10,15].

### 3.2. MI after Chest Trauma

MI can be caused by coronary artery dissection resulting from blunt chest trauma in athletes [19,20]. It is important to consider the potential occurrence of coronary artery damage following blunt thoracic trauma. To assess coronary injuries and determine appropriate treatment, it is recommended to promptly perform an electrocardiogram (ECG) in individuals experiencing chest discomfort or shortness of breath following blunt chest trauma [21].

The occurrence of such a situation is exceptionally uncommon. Prompt recognition and appropriate management, including antiplatelet therapy and emergent interventions, are crucial for the successful treatment of patients. Although such cases are rare, healthcare providers should be aware of the possibility of coronary artery dissection in athletes presenting with acute chest pain. In cases of chest trauma, the left anterior descending artery is the most commonly affected vessel, followed by the right coronary artery, and sometimes the left circumflex artery [19].

### 3.3. Abnormalities of the Coronary Arteries

Congenital abnormalities of the coronary arteries are uncommon, occurring in around 1% of the overall population [22]. It was found that 11% of the causes of SCD in athletes were due to coronary artery anomalies [23]. Another recent study has shown that 48% of cases of SCD related to coronary artery abnormalities occurred during or promptly after physical activity. The most common anomaly, which was identified in 10 out of 31 cases (32%), was the anomalous origin of the right coronary artery from the left coronary sinus [24]. Also, an anomalous left coronary artery can cause severe, potentially fatal MI and SCD. The left coronary artery originating from the opposite sinus of Valsalva with an intramural course is an uncommon congenital heart abnormality that can manifest in early life [25,26]. Symptoms such as recurrent syncope, myocardial ischemia, or ventricular arrhythmias in younger patients should be alarming [26].

Another anomaly that can cause MI is isolated hypoplasia and significant shortening of the coronary artery [22]. Florian-Nikolaus Riede et al. described the case of a 16-year-old athlete who died due to MI. The autopsy revealed a posterolateral MI with a focus on ischemic cardiomyopathy with hypoplasia. The hypoplastic left circumflex artery led to mild hypoplasia of the left ventricle and focal ischemic cardiomyopathy. The MI was hemorrhagic and surrounded by necrotic changes, suggesting a prolonged period of ischemia [22].

Myocardial bridging is a congenital anomaly characterized by the passage of a coronary artery segment through the underlying myocardium [27]. Coronary angiography reveals this condition in approximately 1–3% of patients [28]. Several cases of MI occurring in young athletes associated with the presence of myocardial bridges have been described [28,29,30,31]. This abnormality can lead to the constriction of the artery during systole resulting in a range of cardiovascular issues, such as angina, myocardial ischemia, acute coronary syndrome, left ventricular dysfunction, arrhythmias, and SCD [27]. A rapid increase in heart activity during intense exercise can affect the blood flow through the muscle bridge, which can lead to ischemia and MI by the formation of a thrombus around the myocardial bridge. There are various theories regarding the mechanisms that can lead to ischemia as a result of myocardial bridging, including changes in blood flow, endothelial dysfunction, and arterial narrowing; however, the exact mechanism is still unknown [29]. One of them also includes atherosclerosis developing in the area of the myocardial bridge [27].

It is crucial to acknowledge the significance of non-invasive techniques, like CTCA, in identifying coronary abnormalities, including myocardial bridging; additionally, other non-invasive methods, such as stress echocardiography, can be valuable in detecting these abnormalities, particularly myocardial bridging. In fact, the presence of myocardial bridging can lead to myocardial ischemia following a stress-inducing factor [32].

### 3.4. Coronary Artery Spasm

Recurrent MI in young individuals with no apparent causes and with normal coronary arteries can be attributed to coronary spasms. The autonomic nervous system’s sympathetic activation, platelet aggregation, and the release of thromboxane A2 play significant roles in the pathogenesis of coronary spasms, which can extend along the blood vessel. The coronary spasm causes restriction of blood flow and myocardial hypoxia, which can lead to MI [33]. When coronary artery spasm is identified or strongly suspected during physical exercise, treatment should be initiated with calcium channel blockers and nitrates to reduce the risk of spasm and control symptoms [34].

Coronary artery spasms can also be caused by the consumption of synephrine, which is a dietary supplement commonly used to enhance athletic performance and promote weight loss [35]. Synephrine belongs to the class of sympathetic adrenergic agonists and shares a structural similarity with ephedrine [35]. It functions by activating alpha-1 adrenergic receptors, leading to constriction of blood vessels in the periphery and coronary arteries, as well as inducing hypertension [36]. Cases of MI in young athletes without risk factors or previous symptoms following the ingestion of synephrine have been described in the literature [35,36,37].

### 3.5. Plaque Erosion

Plaque erosion is a significant but often undiagnosed cause of MI. It refers to the erosion of the endothelial layer of an intact fibroatheroma, and it is frequently seen in females, smokers, and younger individuals who have fewer risk factors for cardiovascular disease. The trigger for plaque erosion is not well understood but extreme physical exertion and sympathetic stimulation may play a role [38]. A recent study has found that greater physical activity is associated with the occurrence of plaque erosion [39]. Physical activity more frequently caused plaque erosion than plaque rupture [39]. It can be a cause of both ST-elevation MI and non-ST-elevation MI [38,39]. Another study has found that athletes have a higher prevalence of atherosclerotic plaques compared to sedentary males with similar risk factors; however, the stable nature of these plaques may lower the risk of plaque rupture and sudden myocardial infarction [40]. Optical coherence tomography (OCT) is the preferred imaging method for plaque erosion detection [38]. Finally, in the case of young athletes with very pronounced risk factors, coronary artery disease can be a leading cause of SCD, as demonstrated in the case series of 30 joggers at a mean age of 36 years [41]. Also, depression can have a significant impact on promoting and accelerating the development of atherosclerosis and its associated complications, including plaque rupture and thrombosis, particularly in young patients [42].

### 3.6. Hypercoagulability

Vigorous physical activity may potentially increase the risk of coronary thrombosis due to the transient increase in factor VIII levels [9]. A study has shown that high-intensity physical exercise can provoke changes in the coagulation system, increasing prothrombotic activity, both in patients with coronary artery disease and without [43]. Another study has found that moderate training increases factor vWf levels. Particularly in the male group, the increase was higher compared to the female group [44]. It is believed to be associated with the release of adrenaline and stimulation of β2 adrenergic receptors. It may be necessary to further assess patients’ risk of physical activity, taking into account factors such as body mass index (BMI), gender, ethnicity, pre-existing medical conditions, and hypercoagulability tests. With advances in genetic testing for hypercoagulability becoming more accessible and accurate, regular screening for these thromboembolic disorders as a form of primary prevention before any events occur may be a reasonable approach [9]. It can be especially dangerous in athletes taking anabolic steroids because they have a prothrombotic effect as well [45,46]. Additionally, the level of factor VIII increases with age [9]. Another factor playing a part in the increased hypercoagulability status is dehydration from prolonged exercise, which also increases blood viscosity.

### 3.7. Left Ventricular Hypertrophy

Another potential cause of MI in young athletes is oxygen demand/supply mismatch during strenuous exercise in the case of left ventricular hypertrophy. LVH can be caused by arterial hypertension or HCM. Ehses et al. reported a case of a 16-year-old highly trained athlete who died after suffering ventricular fibrillation during exhaustive physical activity caused by myocardial ischemia [47]. An autopsy revealed concentric myocardial hypertrophy without signs of coronary artery disease. Drug abuse was excluded. A similar situation, luckily with only a small MI, was observed in Case 2 presented in this manuscript. Therefore, screening for undiagnosed hypertension and HCM should be of paramount importance. While concentric hypertrophy is in general more typical for arterial hypertension, an asymmetric one is more characteristic of HCM. Case 2 should also be regarded as a warning sign towards more liberal admission of athletes with low-risk HCM to vigorous and even competitive sports [48]. In case of suspected HCM, it is always worth performing screening of first-degree relatives, preferably with resting ECG and echocardiography. In doubtful cases, genetic testing may also be considered.

### 3.8. MI Induced by Anabolic Steroids

Anabolic steroids are used by athletes, especially by bodybuilders and weightlifters to increase muscle mass [49]. The majority of individuals using anabolic steroids are not associated with professional sports. They take anabolic steroids to enhance physical attractiveness [50,51]. The average age of steroid users is around 30 years old. The use of anabolic steroids increases the risk of SCD in athletes by 6 to 20 times [45]. A few cases of MI associated with the use of anabolic steroids in athletes have been reported [44,45,49,50,51,52,53,54,55]. It has been shown that testosterone therapy for 90 days can increase the risk of MI [54]. Young athletes who abuse anabolic steroids often ignore their symptoms and attribute their symptoms to training. This results in late reporting to the emergency room, with symptoms already present and complications [49].

Anabolic steroids such as exogenous testosterone are associated with nonatherosclerotic coronary artery disease and acute myocardial infarction [52]. There are several possible mechanisms as a cause for MI induced by anabolic steroids: atherogenic, vasospastic, and thrombotic. The acceleration of atherosclerosis is caused by reducing the level of HDL cholesterol and increasing the level of LDL cholesterol in the blood. Coronary vasospasm is caused by the inhibition of guanylate cyclase. Anabolic steroids also enhance platelet aggregation by promoting the production of thromboxane A2, a powerful platelet aggregator, while simultaneously reducing the production of prostacyclin, which inhibits platelet aggregation [44,45]. The use of anabolic steroids can also lead to other cardiovascular issues, including hypertension, left ventricular hypertrophy, impaired diastolic filling, or polycythemia [53].

There is a potential association between substance use or abuse and the occurrence of MINOCA. A recent study has found that in the MINOCA group, 43% of individuals had a history of substance use or abuse. The substances reported in the MINOCA group included alcohol, marijuana, cocaine, heroin, methadone, anabolic steroids, amphetamine, and energy drinks. It is worth noting that eight individuals (50%) in the MINOCA group reported using two or more of these substances [56].

## 4. Considerations for Clinical Practice

The incidence of acute MI with non-obstructive coronary arteries in young patients is rising, surpassing the prevalence of traditional obstructive MI. The significant role of CMR has been observed in diagnosing and managing such cases [57]. Enabling appropriate screening to identify athletes at risk is crucial. For this purpose, it is important to take a thorough medical history, particularly considering the family history, smoking, obesity, alcohol consumption, and the use of anabolic steroids, cocaine, and marijuana [58]. Additionally, it is necessary to regularly conduct tests such as ECG or echocardiographic examination of the heart in search of coronary artery anomalies or left ventricular hypertrophy. Laboratory analyses can reveal polycythemia and lipid profile abnormalities related to the use of steroids. In individuals with abnormalities, the diagnostic workup should be expanded depending on the suspected cause. This allows effective management and treatment, reducing the risk of complications and preventing myocardial infarction in young athletes—according to European Society of Cardiology guidelines on cardiovascular disease prevention in clinical practice—which should not be overlooked in young, healthy-looking, fit athletes [59]. Moreover, it is important not to underestimate the symptoms, to implement prompt treatment, and not to engage in strenuous physical activities beyond the current state of physical preparation or possibilities. Symptoms such as acute chest pain and recurrent syncope during or immediately after physical exertion should be always alarming.

## 5. Conclusions

The occurrence of MI in young athletes is not common; however, due to the young age of the patients and the potential absence of risk factors, it always generates significant interest. The main causes described in the literature on young athletes include coronary artery dissection, coronary artery spasm, chest trauma, abnormalities of the coronary arteries, plaque erosion, left ventricular hypertrophy, and hypercoagulability. Additionally, the use of steroid hormones is associated with an increased risk of MI. Understanding the mechanisms of MI occurrence in this population is important for identifying athletes at risk and preventing acute MI and its complications. The available literature describes a limited number of scientific studies and reported cases of MI in young athletes; therefore, conducting further research on this topic is necessary.

## Figures and Tables

**Figure 1 diagnostics-13-02473-f001:**
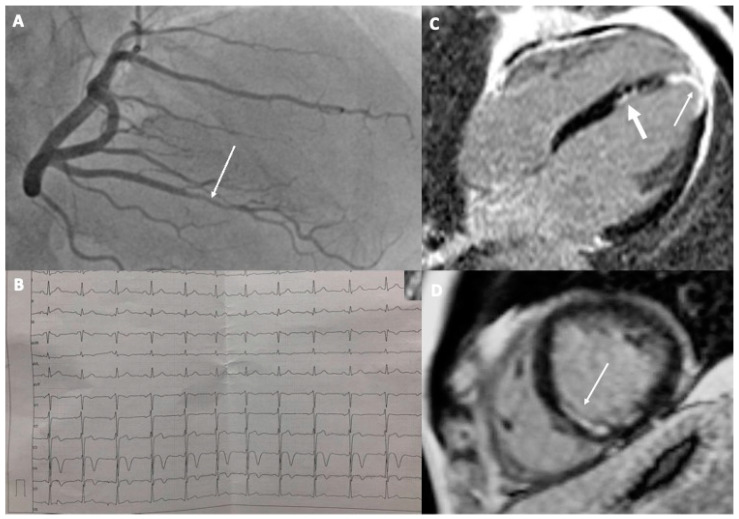
A 37-year-old athlete with recurrent MI: (**A**) Coronary angiogram of the right coronary artery with visible lower opacification in one of the branches (arrow); (**B**) Electrocardiogram with visible signs of ischemia in the form of T-wave inversion in V4 and V5; (**C**,**D**) Cardiac magnetic resonance images after second MI showing subendocardial scar in the interventricular septum and in the apex typical for myocardial infarction (thin arrows). Previous MI is visible in the interventricular septum ((**C**)—thicker arrow).

**Figure 2 diagnostics-13-02473-f002:**
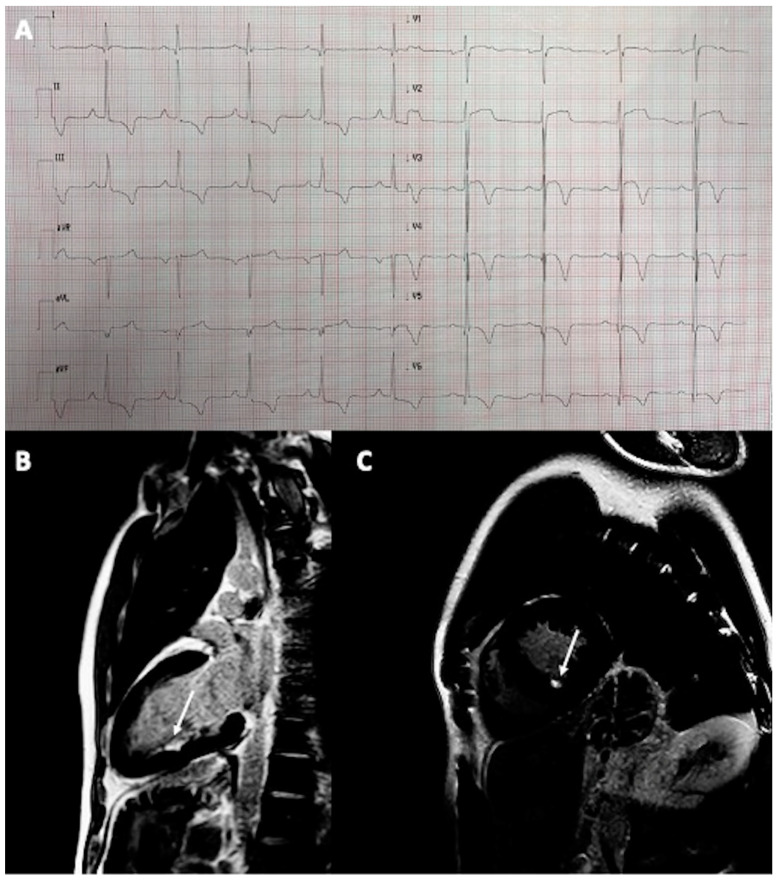
A 36-year-old male athlete with HCM and MI: (**A**) Electrocardiogram showing signs of LVH described above; (**B**,**C**) Cardiac magnetic resonance demonstrating small subendocardial scar in the interventricular septum (arrows).

**Figure 3 diagnostics-13-02473-f003:**
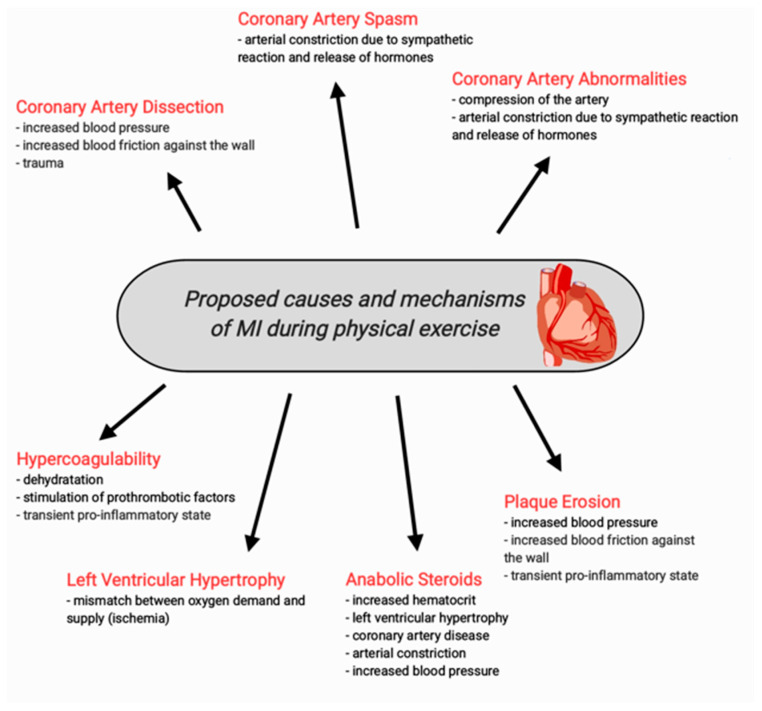
Proposed causes and mechanisms of MI during physical activity in young athletes.

## Data Availability

Not applicable.

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
