# Peer review of "Myocardial Infarction in Young Athletes"

_diagnostics, 2023, doi:10.3390/diagnostics13152473_

Round 1

Reviewer 1 Report

Dear Authors,

I appreciate your efforts in carrying out this research on an interesting topic. However, I have a few concerns that need to be addressed.

1.       Title: The case presented in your manuscript falls under the “middle-age” bracket. However, the rest of the manuscript deals with young age group athletes which according to the literature should be up to late twenties. Kindly clarify.  

2.       Add section: interesting topic, but I feel a section on preventive approaches including pharmacological interventions can be added.    

Best Wishes

Minor English revision is required. 

Author Response

I appreciate your efforts in carrying out this research on an interesting topic. However, I have a few concerns that need to be addressed.

  1. Title:The case presented in your manuscript falls under the “middle-age” bracket. However, the rest of the manuscript deals with young age group athletes which according to the literature should be up to late twenties. Kindly clarify.  

According to the criteria used in the ESC guidelines for sports cardiology published in 2020, young athletes are considered up to the age of 35 years, and after that, they are classified as veteran-athletes. A margin of +/-5 years is often used to define this population.

  1. Add section:interesting topic, but I feel a section on preventive approaches including pharmacological interventions can be added.    

We have added the following sentence to the future directions section, renamed Considerations for clinical practice: “This allows effective management and treatment, reducing the risk of complications and preventing myocardial infarction in young athletes according to European Society of Cardiology guidelines on cardiovascular disease prevention in clinical practice, which should not be overlooked in young, healthy-looking, fit athletes [new ref].”

[new ref] Visseren, Frank LJ, et al. “2021 ESC Guidelines on cardiovascular disease prevention in clinical practice: Developed by the Task Force for cardiovascular disease prevention in clinical practice with representatives of the European Society of Cardiology and 12 medical societies With the special contribution of the European Association of Preventive Cardiology (EAPC)”. European Heart Journal, vol. 42, 34 (2021): 3227-3337. https://doi.org/10.1093/eurheartj/ehab484

Reviewer 2 Report

This paper present two distinct case of MI in athletes and a literature review.

The manuscript is well written.

I suggest also to elaborate on the risk of sudden cardiac death (SCD), in particular related to the use of abuse of substances , which is not uncommon among young SCD victims, especially if the cause of death is myocardial infarction, even in the absence of obstructive coronary disease (consider: Arterioscler Thromb Vasc Bio2023 May;43(5):787-792.)

minor issues

Author Response

I suggest also to elaborate on the risk of sudden cardiac death (SCD), in particular related to the use of abuse of substances , which is not uncommon among young SCD victims, especially if the cause of death is myocardial infarction, even in the absence of obstructive coronary disease (consider: Arterioscler Thromb Vasc Bio. 2023 May;43(5):787-792.) 

Thank you for that comment. We added this information: ,,There is a potential association between substance use or abuse and the occurrence of MINOCA. A recent study has found that in the MINOCA group, 43% of individuals had a history of substance use or abuse. The substances reported in the MINOCA group included alcohol, marijuana, cocaine, heroin, methadone, anabolic steroids, amphetamine, and energy drinks. It is worth noting that 8 individuals (50%) in the MINOCA group reported using two or more of these substances [new ref].’’   

[new ref] Ciliberti, Giuseppe et al. “Coronary Artery Dissection and Myocardial Infarction With Nonobstructed Coronary Arteries: Insights From a UK Nationwide Autopsy-Based Registry-Brief Report.” Arteriosclerosis, thrombosis, and vascular biology vol. 43,5 (2023): 787-792. doi:10.1161/ATVBAHA.122.318401

Reviewer 3 Report

The authors have presented a captivating review on Myocardial Infarction in young athletes, offering valuable insights into this topic. The manuscript is well-structured and written in an engaging manner, making it an enjoyable read for both researchers and clinicians interested in this area. One notable strength of the review is the inclusion of two intriguing case reports. These case reports offer valuable context and highlight the importance of early recognition, diagnosis, and appropriate management of Myocardial Infarction in young athletes.

Some Major Issues: 

Please consider improving the CMR part of Figure 2 (panel B) and Figure 1 (panel D).

Paragraph 3.1: Consider also the role of CCTA for SCAD detection.

Paragraph 3.3 In addition to the points mentioned earlier, it is important to consider the role of non-invasive methodologies, such as coronary computed tomography angiography (CCTA), in the detection of coronary abnormalities (CA) and specifically myocardial bridging. Other non-invasive methods, such as stress echocardiography, could be helpful for the detection of CA, especially myocardial bridging. In fact, the presence of myocardial bridging can determine myocardial ischemia after a stressor (reference: https://doi.org/10.1016/j.echo.2023.06.008).

Paragraph 3.5: Depression can have a significant impact on promoting and accelerating the development of atherosclerosis and its associated complications, including plaque rupture and thrombosis, particularly in young patients (reference: PMID: 23241128).

In the introduction section or in the “Future Directions” Section, it is important to address the increasing prevalence of acute myocardial infarction (AMI) with non-obstructive coronary artery in young patients, which has been observed to be greater than the prevalence of classic obstructive AMI and the crucial role of CMR in these cases (reference: 10.1016/j.jcmg.2022.12.029). 

Finally, consider also adding a brief specific paragraph regarding the role of other etiologies related to SCD in young athletes - other cardiomyopathies (eg ARVC), myocarditis, valvulopathies (such as MVP) and channelopathies (eg LQT, CPVT)  (reference PMC4969030, https://doi.org/10.1016/j.jacc.2023.01.041) - that must be considered in the differential diagnosis.

Minor issue

Coronary artery disease, with or without acute MI, is identified as the most common underlying pathology in all three groups [7]. Which groups?

Minor English revision is required. Also, I suggest double-checking the abbreviations.

Author Response

The authors have presented a captivating review on Myocardial Infarction in young athletes, offering valuable insights into this topic. The manuscript is well-structured and written in an engaging manner, making it an enjoyable read for both researchers and clinicians interested in this area. One notable strength of the review is the inclusion of two intriguing case reports. These case reports offer valuable context and highlight the importance of early recognition, diagnosis, and appropriate management of Myocardial Infarction in young athletes.

Some Major Issues: 

Please consider improving the CMR part of Figure 2 (panel B) and Figure 1 (panel D).

            The quality of the mentioned figure 1 and 2 panels has been improved.

Paragraph 3.1: Consider also the role of CCTA for SCAD detection.

Thank you for that comment. We have added information about the role of CCTA in the diagnosis of SCAD: ,,Thanks to advances in technology, CCTA plays an increasingly important role in the diagnosis of SCAD. However, the precise criteria for the diagnosis of SCAD by CCTA, as well as the sensitivity and specificity of such a test, have not yet been established [new ref].’’

[new ref] Aslam, Anum et al. “Spontaneous Coronary Artery Dissection: An Underdiagnosed Clinical Entity-A Primer for Cardiac Imagers.” Radiographics : a review publication of the Radiological Society of North America, Inc vol. 41,7 (2021): 1897-1915. doi:10.1148/rg.2021210062

[new ref] Pergola, Valeria et al. “Spontaneous coronary artery dissection: the emerging role of coronary computed tomography.” European heart journal. Cardiovascular Imaging vol. 24,7 (2023): 839-850. doi:10.1093/ehjci/jead060

Paragraph 3.3 In addition to the points mentioned earlier, it is important to consider the role of non-invasive methodologies, such as coronary computed tomography angiography (CCTA), in the detection of coronary abnormalities (CA) and specifically myocardial bridging. Other non-invasive methods, such as stress echocardiography, could be helpful for the detection of CA, especially myocardial bridging. In fact, the presence of myocardial bridging can determine myocardial ischemia after a stressor (reference: https://doi.org/10.1016/j.echo.2023.06.008).

We have added the following information: ,,It is crucial to acknowledge the significance of non-invasive techniques, like CCTA, in identifying coronary abnormalities, including myocardial bridging. Additionally, other non-invasive methods, such as stress echocardiography, can be valuable in detecting these abnormalities, particularly myocardial bridging. In fact, the presence of myocardial bridging can lead to myocardial ischemia following a stress-inducing factor [new ref.]’’

[new ref] Guerra, Emiliano et al. “Contrast stress-echocardiography findings in myocardial bridging compared to normal coronary course, with and without coronary artery disease.” Journal of the American Society of Echocardiography : official publication of the American Society of Echocardiography, S0894-7317(23)00318-8. 23 Jun. 2023, doi:10.1016/j.echo.2023.06.008

Paragraph 3.5: Depression can have a significant impact on promoting and accelerating the development of atherosclerosis and its associated complications, including plaque rupture and thrombosis, particularly in young patients (reference: PMID: 23241128).

We have added this information to our study: „ Also, depression can have a significant impact on promoting and accelerating the development of atherosclerosis and its associated complications, including plaque rupture and thrombosis, particularly in young patients [new ref].”

[new ref] Pizzi, C et al. “Pathophysiological mechanisms linking depression and atherosclerosis: an overview.” Journal of biological regulators and homeostatic agents vol. 26,4 (2012): 775-82.

In the introduction section or in the “Future Directions” Section, it is important to address the increasing prevalence of acute myocardial infarction (AMI) with non-obstructive coronary artery in young patients, which has been observed to be greater than the prevalence of classic obstructive AMI and the crucial role of CMR in these cases (reference: 10.1016/j.jcmg.2022.12.029). 

We have added this information to our manuscript: “The incidence of acute MI with non-obstructive coronary arteries in young patients is rising, surpassing the prevalence of traditional obstructive MI. The significant role of cardiac magnetic resonance imaging (CMR) has been observed in diagnosing and managing such cases [new ref.].”

[new ref.] Mileva, Niya et al. “Diagnostic and Prognostic Role of Cardiac Magnetic Resonance in MINOCA: Systematic Review and Meta-Analysis.” JACC. Cardiovascular imaging vol. 16,3 (2023): 376-389. doi:10.1016/j.jcmg.2022.12.029

Finally, consider also adding a brief specific paragraph regarding the role of other etiologies related to SCD in young athletes - other cardiomyopathies (eg ARVC), myocarditis, valvulopathies (such as MVP) and channelopathies (eg LQT, CPVT)  (reference PMC4969030, https://doi.org/10.1016/j.jacc.2023.01.041) - that must be considered in the differential diagnosis.

Our study was focused on myocardial infarction in young athletes, which is a rarely discussed topic in the literature. We did not want to focus on all potential causes of SCD in young athletes, which are of numerous and discussed in detail in other studies, including sports cardiology guidelines as mentioned by the reviewer. We wanted to limit the introduction and discussion to the main topic to deliver a stronger message on the main topic.

Minor issue

Coronary artery disease, with or without acute MI, is identified as the most common underlying pathology in all three groups [7]. Which groups?

We have added precise clarification in our study regarding which groups are being referred to: ,,in the group of individuals not associated with sports, involved in recreational or competitive sports.’’

Minor English revision is required. Also, I suggest double-checking the abbreviations.

            We have addressed these comments.

Reviewer 4 Report

The topic of this paper is interesting, but both case report #2 and discussion of HCM as cause of MI in young athletes, need to be improved.

First, in the diagnostic evaluation of the pt #2, presenting with asymmetric LV hypertrophy, I suppose that hypertension and any other possible cause of secondary cardiac hypertrophy (e.g., infiltrative disorders, endocrine diseases, drugs etc.- see ESC guidelines for diagnosis and treatment of HCM, 2014) have been excluded. Please, specify this point; the Authors could also comment on the different meaning of asymmetric vs. concentric LV hypertrophy. Moreover, the patient did not have any known familial history of HCM, but it could be appropriate to mention and explain (in Discussion, sub-section 3.7) when genetic testing are appropriate for HCM diagnostic work-up, such as whether and when non-invasive and low-cost, widely used and easily accessible tests (i.e., ECG and Echocardiogram) must be performed also in (or at least proposed) siblings/relatives of the patients with HCM.

Finally, the English has to be revised throughout the text.

The English language needs revision; it is quite easy to understand the meaning of the sentences, but there are a few (minor) errors throughout the text.

Example: "It can cause no symptoms... " to be corrected as "It can cause no symptom ..."

Author Response

The topic of this paper is interesting, but both case report #2 and discussion of HCM as cause of MI in young athletes, need to be improved.

First, in the diagnostic evaluation of the pt #2, presenting with asymmetric LV hypertrophy, I suppose that hypertension and any other possible cause of secondary cardiac hypertrophy (e.g., infiltrative disorders, endocrine diseases, drugs etc.- see ESC guidelines for diagnosis and treatment of HCM, 2014) have been excluded. Please, specify this point;

We have added this information to our study: “Other possible causes of secondary cardiac hypertrophy have been excluded as per guidelines on the diagnosis and treatment of hypertrophic cardiomyopathy [new ref].”

[new ref] Elliott, Perry M, et al. “2014 ESC Guidelines on diagnosis and management of hypertrophic cardiomyopathy: the Task Force for the Diagnosis and Management of Hypertrophic Cardiomyopathy of the European Society of Cardiology (ESC)” Eur Heart J vol 35, 39 (2014); 2733-2779.

The Authors could also comment on the different meaning of asymmetric vs. concentric LV hypertrophy. Moreover, the patient did not have any known familial history of HCM, but it could be appropriate to mention and explain (in Discussion, sub-section 3.7) when genetic testing are appropriate for HCM diagnostic work-up, such as whether and when non-invasive and low-cost, widely used and easily accessible tests (i.e., ECG and Echocardiogram) must be performed also in (or at least proposed) siblings/relatives of the patients with HCM.

Thank you for your comment. We have added the following sentences to sub-section 3.7: “While concentric hypertrophy is in general more typical for arterial hypertension, asymmetric one is more characteristic of HCM. … In case of suspected HCM, it is always worth performing screening of 1st-degree relatives preferably with resting ECG and echocardiography. In doubtful cases, genetic testing may also be considered”.

Finally, the English has to be revised throughout the text. The English language needs revision; it is quite easy to understand the meaning of the sentences, but there are a few (minor) errors throughout the text. Example: "It can cause no symptoms... " to be corrected as "It can cause no symptom ..."

We have addressed this issue.

Round 2

Reviewer 3 Report

Congratulations to the authors. The manuscript is much improved and all my questions have been answered. 

The quality of English is good.